# Precautions during Direct Oral Anticoagulant Introduction in Gynecologic Malignancies: A Single-Center Retrospective Cohort Study

**DOI:** 10.3390/cancers15041132

**Published:** 2023-02-10

**Authors:** Takanori Shimizu, Noriyuki Iwama, Hideki Tokunaga, Shun Endo, Shuko Miyahara, Asami Toki, Zen Watanabe, Junko Minato, Chiaki Hashimoto, Masumi Ishibashi, Shogo Shigeta, Muneaki Shimada, Nobuo Yaegashi

**Affiliations:** 1Department of Obstetrics and Gynecology, Tohoku University Hospital, 1-1, Seiryomachi, Sendai 980-8574, Japan; 2Tohoku Medical Megabank Organization, Tohoku University, 2-1, Seiryomachi, Sendai 980-8573, Japan; 3Division of Gynecology, Miyagi Cancer Center, 47-1, Nodayama, Medeshima-Shiode, Natori 981-1293, Japan

**Keywords:** D-dimer, direct oral anticoagulants, gynecologic cancer, risk factors, venous thromboembolism

## Abstract

**Simple Summary:**

This study aimed to elucidate the risk factors for venous thromboembolism (VTE) recurrence/exacerbation or a change from a direct oral anticoagulant (DOAC) to another anticoagulant in patients with gynecologic cancer using DOACs. Our investigation reveals that ovarian clear cell carcinoma, pulmonary embolism (PE) or proximal deep vein thrombosis without PE, and elevated D-dimer levels at VTE diagnosis are associated with increased odds of a primary outcome in this population. The findings of this study will provide clinicians with a scientific basis to facilitate the identification of high-risk patients and ensure prompt treatment and careful follow-up after DOAC initiation for the early recognition of treatment failure.

**Abstract:**

The risk factors for venous thromboembolism (VTE) recurrence/exacerbation or a change from a direct oral anticoagulant (DOAC) to another anticoagulant in patients with gynecologic cancer using DOACs have not been thoroughly elucidated. Here, we aimed to investigate the risk factors for a composite primary outcome, including VTE recurrence/exacerbation, or a change from a DOAC to another anticoagulant, in this population. A total of 63 patients were analyzed. Risk factors for a primary outcome within 2 years after DOAC initiation were investigated using multiple logistic regression analysis. Among the 63 patients, 10 developed a primary outcome. Clear cell carcinoma of the ovary (adjusted odds ratio (aOR), 18.9; 95% confidence interval (CI), 2.25–350.74), pulmonary embolism (PE) or proximal deep vein thrombosis without PE (aOR, 55.6; 95% CI, 3.29–11,774.66), and D-dimer levels in the third tertile (≥7.6 μg/dL) when VTE was first diagnosed (aOR, 6.37; 95% CI, 1.17–66.61) were associated with increased odds of a primary outcome in patients with gynecologic cancer using DOACs. Patients with one or more risk factors for a primary outcome require careful follow-up after DOAC initiation for the early recognition of treatment failure.

## 1. Introduction

Venous thromboembolism (VTE), including deep vein thrombosis (DVT) and pulmonary embolism (PE), is a common complication in patients with malignant tumors and is associated with a poor prognosis [1]. Conventionally, unfractionated heparin (UFH) and warfarin potassium have been used as initial treatments for VTE in the acute phase [2]. However, UFH requires regular monitoring and is complicated to administer since it is not administered orally [3]. Additionally, the narrow therapeutic index of warfarin complicates the need for frequent international standardization ratio monitoring to prevent bleeding complications and maintain therapeutic efficacy [4]. Furthermore, many foods and drugs—not just alcohol—interact with warfarin metabolism; therefore, with warfarin administration, it is necessary to pay attention to daily life [5]. Moreover, genetic variation in the metabolism of warfarin is known to affect its efficacy [5].

In contrast, a direct oral anticoagulant (DOAC) has several advantages, such as no requirement for regular blood monitoring or dose adjustment as well as possessing no drug–food interactions [6]. The DOACs available in Japan include factor Xa and factor IIa inhibitors. In endogenous coagulation, Xa converts prothrombin to thrombin and activates coagulation factors that form a sequential cascade, whereas in extrinsic coagulation, tissue factors are expressed in subepithelial fibroblasts, factor VII is activated in the bloodstream, and calcium ions form a complex. This complex activates the formation of insoluble cross-linked fibrin [7]. When coagulation factor X is activated, a large amount of coagulation factor Xa is induced from the coagulation cascade in both the extrinsic and endogenous systems. Coagulation factor Xa cleaves prothrombin to generate thrombin (IIa), leading to thrombus formation. Therefore, directly inhibiting coagulation factor Xa prevents thrombin formation, leading to an anticoagulant effect [8]. In Japan, the DOACs that are approved for the treatment of VTE are Xa inhibitors.

Several large-scale international clinical trials, including the Hokusai-VTE, EINSTEIN, and AMPLIFY trials, report that DOACs have a lower risk of bleeding and recurrence of VTE than conventional anticoagulants, such as low-molecular-weight heparin (LMWH), UFH, and vitamin K antagonists (i.e., warfarin or acenocoumarol) [6,7,9]. In addition, DOACs are not inferior to conventional anticoagulants in terms of therapeutic efficacy and complications [6,7,9].

The American Society of Clinical Oncology (ASCO) and the National Comprehensive Cancer Network guidelines recommend the use of LMWH, UFH, fondaparinux, and DOACs, including apixaban, dabigatran, rivaroxaban, and edoxaban, for anticoagulant therapy of VTE in patients with cancer [9]. In the original Japanese version, the 2020 Japan Society of Gynecologic Oncology guidelines for treating ovarian, fallopian tube, and primary peritoneal cancers indicate that ovarian cancer itself is a risk factor for VTE and recommend perioperative management according to the ASCO guidelines [10]. LMWH, a standard VTE treatment in Europe and the United States, cannot be used in Japan to prevent VTE due to health insurance approval issues. Additionally, self-injection UFH is covered by health insurance for thrombosis prevention in patients with a history of VTE only if there is no alternative. Therefore, DOACs are likely to be used more frequently in Japan. Since 2014, in Japan, DOACs, including edoxaban, rivaroxaban, and apixaban, have been approved to treat VTE and prevent its recurrence. Moreover, the latest guidelines for diagnosing, treating, and preventing VTE also recommend using DOACs [2]. However, recurrence of VTE occurs regardless of the use of DOACs [6,7,9]. It is not uncommon for patients to require a change from one anticoagulant to another when using a DOAC [11]. Since D-dimer levels may reflect the efficiency of anticoagulants, selected anticoagulants may be changed with an increase in D-dimer levels [12].

Since gynecological cancer is a risk factor for VTE in cancer-bearing patients [13], information regarding the risk factors for VTE recurrence/exacerbation in patients with gynecologic cancer using DOACs may be useful for detecting high-risk patients, as well as for early evaluation and interventions. However, the risk factors for VTE recurrence/exacerbation or a change from a DOAC to another anticoagulant have not been well investigated. Therefore, this study aims to elucidate these risk factors in patients with gynecologic cancer who are using DOACs.

## 2. Materials and Methods

### 2.1. Study Design and Participants

This was a retrospective cohort study conducted at the Department of Gynecology at Tohoku University Hospital (Sendai, Japan), one of the high-volume cancer centers in Japan, where several clinical studies on gynecologic malignancies have been conducted [14,15]. We collected data from patients who met all of the following registration criteria: (1) patients who first visited the Department of Gynecology of Tohoku University Hospital between January 2016 and October 2019; (2) patients newly diagnosed with gynecologic malignancies, including cervical, endometrial, and ovarian cancers; and (3) patients who had venous blood drawn and recorded D-dimer levels for screening VTE from when they first visited the Department of Gynecology of Tohoku University Hospital.

Ultrasonography of the lower limb or contrast-enhanced computed tomography (CT) was performed to diagnose VTE when patients presented with symptoms, including D-dimer levels of ≥1.5 μg/dL, imbalance in the lower leg circumference, and respiratory distress. Following VTE diagnosis, a cardiologist or vascular surgeon decided on the treatment. Anticoagulants were used in reference to the Japanese guidelines for treatment of VTE [2]. We planned to include patients treated with DOACs for VTE in our analyses. In addition, we planned to collect data over a 2-year period from the initiation of a DOAC. This study was approved by the Ethics Review Board of Tohoku University Hospital (approval number: 2020-1-835). The Ethics Review Board decided that informed consent from the study subjects was not required because this study did not use samples taken from the human body and is an academic study. Therefore, we applied the opt-out method to obtain consent; a poster was used to disclose information regarding the implementation of the survey, including the purpose of the study.

### 2.2. Primary Outcome

The primary outcome in this study was a composite outcome, including the recurrence/exacerbation of VTE or a change from a DOAC to another drug within 2 years after DOAC initiation. Therefore, we investigated medical records to collect information on the recurrence of VTE within 2 years after the initiation of a DOAC. Recurrence/exacerbation of VTE was defined as the appearance of new thrombi or exacerbation of a thrombosis, confirmed by ultrasonography or CT. A cardiologist or vascular surgeon at our hospital regarded a thrombus as poorly controlled based on an increase in D-dimer or fibrin monomer levels. Upon this decision, even if no clinical symptoms of VTE recurrence were present, the DOAC was changed to other anticoagulants, including VKA, different types of DOACs, or parenteral anticoagulants (intravenous or subcutaneous heparin).

### 2.3. Other Variables

Age, body mass index (BMI), performance status (PS), hypertension, diabetes mellitus, diagnosis of gynecologic malignancies, International Federation of Gynecology and Obstetrics (FIGO) stage, final prognosis, and continuation status of DOAC were collected. The FIGO 2008 staging was adopted for cervical and endometrial cancers, and the FIGO 2014 staging was adopted for ovarian cancer. In addition, we collected data on the situation (before or after treatment) and location of blood clots when VTE was first diagnosed; location of blood clots when VTE recurrence was diagnosed; blood sampling data, including white blood cell (WBC) and platelet counts and hemoglobin, D-dimer, and fibrin monomer levels, when VTE was first diagnosed; and DOAC-related adverse events. The location of blood clots when VTE was first diagnosed was classified into “PE or proximal DVT without PE” and “isolated distal DVT”. WBC, platelet counts, and hemoglobin, D-dimer, and fibrin monomer levels when VTE was first diagnosed were divided into tertiles.

### 2.4. Statistical Analyses

All statistical analyses were performed using R version 4.1.1 [16]. Continuous and categorical variables were expressed as median (interquartile range (IQR)) and number (percentage), respectively.

We first applied a univariate logistic regression model to explore candidate explanatory variables significantly associated with the primary outcome, because there was a small number of subjects analyzed in this study. There was no strong multicollinearity among explanatory variables, with each two-sided *p*-value < 0.10 in a univariate logistic regression model, which we confirmed using the variance inflation factor of a general linear model. Next, all explanatory variables with two-sided *p*-values < 0.10 in the univariate logistic regression model were included in a multiple logistic regression model. As described in the “Risk factors for a primary outcome” section, clear cell carcinoma of the ovary, PE or proximal DVT without PE when VTE was first diagnosed, and D-dimer levels in the third tertile (≥7.6 μg/dL) when VTE was first diagnosed were included in the multiple logistic regression model. We used Firth’s logistic regression model when complete or quasi-complete separation occurred in the univariate and/or multiple logistic regression models, using the maximum likelihood method [17].

As an additional analysis, we estimated the cumulative incidence function (CIF) for a primary outcome among patients in each explanatory variable used in the multiple logistic regression model. Gray’s test was performed to compare CIFs for a primary outcome between each explanatory variable [18]. In addition, the difference in cause-specific restricted mean time lost (RMTL) among patients in each explanatory variable used in the multiple logistic regression model, considering competing risks, was evaluated using the method proposed by Conner and Trinquart [19]. Cause-specific RMTL is the area under the CIF and is interpreted as the time lost due to each event. The difference in cause-specific RMTL among patients in each explanatory variable is equal to the difference in the area under the CIF. Therefore, the difference in cause-specific RMTL indicates the difference in the lost time due to each explanatory variable in the model. In this study, cause-specific RMTL for a primary outcome was reported. For example, a positive value of the difference in cause-specific RMTL between the third tertile of D-dimer levels (≥7.6 μg/dL) and the first or second tertile of D-dimer levels (<7.6 μg/dL) signifies that patients within the third tertile had developed a primary outcome earlier than those within the first or second tertile, when the latter was set as a reference category. In cause-specific RMTL analysis, the truncation time was set as 730 days, equal to the observation period (i.e., 2 years after DOAC initiation). A two-sided *p*-value < 0.05 was considered statistically significant when evaluating Gray’s test and the difference in cause-specific RMTL among patients in each explanatory variable.

## 3. Results

### 3.1. Characteristics of Study Participants

Figure 1 shows the flowchart of the selection process used in this study. A total of 623 patients met all the registration criteria. Overall, 530 patients who were not initially diagnosed with VTE were excluded. Moreover, 30 patients, who were prescribed anticoagulants other than a DOAC for VTE, including unfractionated heparin or vitamin K antagonists, were excluded. A final count of 63 patients were included. Table 1 shows the characteristics of the participants. Approximately half of the patients had ovarian cancer. In this study, we followed the Japanese guidelines for treatment of gynecological malignancies. Primary treatments for cervical cancer comprise surgery for early stage and radiation therapy for local advanced tumors. Surgery is the first step in the treatment of endometrial cancer, excluding patients with unresectable aggressive tumors or with poor performance status. In Japan, chemotherapy without radiation represents the most popular adjuvant therapy for endometrial cancer. Table 2 shows laboratory variables of those patients. When VTE was first diagnosed, the median (IQR) increases in D-dimer and fibrin monomer levels were 5.1 (IQR, 2.9–9.0) μg/dL and 6.4 (IQR, 3.2–43.9) μg/dL, respectively. The number (percentage) for primary outcome (i.e., recurrence of VTE or change from DOAC to other anticoagulants) was 10 (16%).

### 3.2. Clinical Characteristics of Patients Who Developed VTE Recurrence/exacerbation

Table 3 shows the clinical characteristics of five patients who developed VTE recurrence/exacerbation.

The median (range) of age when a DOAC was initiated was 57 (49–73) years. Additionally, the median (range) of BMI was 21.4 (19–27.5) kg/m^2^, PS was 0 in 3/5 patients, and all patients had no critical medical history.

Ovarian cancer was diagnosed in four patients, among which three suffered clear cell carcinoma. Additionally, FIGO stage was ≥III in four patients.

All cases were diagnosed with VTE for the first time before the first treatment. When VTE was first diagnosed, no cases had a distally located thrombus; therefore, thrombi were noted more proximally in all cases. The median (range) of D-dimer levels when VTE was first diagnosed was 6.6 (2.3–36.9) μg/dL.

As summarized in Table 3, four out of the five patients underwent surgery as a primary treatment, and one had radiation therapy. VTE recurred after the first treatment in four patients. DOACs prescribed were Apixaban and Edoxaban for two and three patients, respectively. The duration of DOAC until recurrence/exacerbation of VTE was divided into two groups (i.e., 1 month or shorter and 1 year or longer). The locations of blood clots upon VTE recurrence included stroke and PE (two patients suffered either a stroke or a PE). Patients with cerebral infarction or PE were cancer-bearing and showed progression of the disease due to chemotherapy resistance or incurable difficulty. During the observation period, four patients died due to primary disease.

### 3.3. Clinical Characteristics of Patients Who Changed from DOACs to Other Anticoagulants

Table 4 shows the clinical characteristics of five patients who changed from DOACs to other anticoagulants according to a cardiologist or vascular surgeon’s decision at our hospital and were deemed as having a poorly controlled thrombus based on an increase in D-dimer or fibrin monomer levels. The median (range) age when a DOAC was initiated was 63 (39–70) years, and the median (range) BMI was 24 (21.1–28.4) kg/m^2^. Additionally, PS was 0 in three of the five patients, and all cases had no critical medical history.

Ovarian cancer was diagnosed in all patients. We confirmed a pathological diagnosis of serous carcinoma in three patients, endometrioid carcinoma in one patient, and clear cell carcinoma in one patient. Additionally, FIGO stage was III or IV in all patients.

All patients were diagnosed with VTE for the first time before the first treatment. When VTE was first diagnosed, no cases with a distally located thrombus were observed. The median (range) of D-dimer levels when VTE was first diagnosed was 15.5 (14.8–34.0) μg/dL. Additionally, for four out of the five patients, the initial treatment was neoadjuvant chemotherapy. A change from a DOAC to another drug occurred before the first treatment in two patients and after the first treatment in three patients. As an index for changing DOACs, fibrin monomer levels were evaluated in four patients, and the median (range) of fibrin monomer increase when DOACs were replaced with other anticoagulants was 31.1 (17.6–34.0) μg/dL. Within a week, three patients were shifted from the initial DOAC to another drug. During the observation period, three patients had disease-free survival, one patient had cancer-bearing survival, and one patient died due to the primary disease.

### 3.4. Risk Factors for a Primary Outcome

Table 5 shows the univariate logistic regression model results. *p*-values for ovarian cancer, clear cell carcinoma of the ovary, PE or proximal DVT without PE when VTE was first diagnosed, and D-dimer levels in the third tertile (≥7.6 μg/dL) when VTE was first diagnosed were <0.10 in the univariate logistic regression model. Although ovarian cancer was associated with increased odds of a primary outcome in the univariate logistic regression model (*p*-value = 0.04), clear cell carcinoma of the ovary was included in the multiple logistic regression model because previous studies indicated that it was associated with a greater risk of disease-related VTE than other types of epithelial ovarian cancer [20,21,22]. Therefore, these three explanatory variables (i.e., clear cell carcinoma of the ovary, PE or proximal DVT without PE when VTE was first diagnosed, and D-dimer levels in the third tertile (≥7.6 μg/dL) when VTE was first diagnosed) were included in the multiple logistic regression model.

Table 6 shows the results of the multiple logistic regression model. The adjusted odds ratios were 18.9 (95% CI, 2.25–350.74) for clear cell carcinoma of the ovary, 55.6 (95% CI, 3.29–11,774.66) for PE or proximal DVT without PE when VTE was first diagnosed, and 6.37 (95% CI, 1.17–66.61) for the third tertile of D-dimer levels (≥7.6 μg/dL) when VTE was first diagnosed.

## 4. Discussion

This study showed that clear cell carcinoma of the ovary, PE or proximal DVT without PE when VTE was first diagnosed, and D-dimer levels when VTE was first diagnosed were risk factors for a primary outcome (i.e., VTE recurrence/exacerbation or a change from a DOAC to another drug) in patients with gynecologic cancer using a DOAC.

The findings of our study are in line with those of previous studies. For example, other studies have also shown that PE or proximal DVT without PE is a risk factor for VTE recurrence [23] and proximal DVT has a higher risk of VTE recurrence than distal DVT [24]. However, Nutescu et al. reported that proximal and distal DVT without PE had a similar risk for VTE recurrence in patients with cancer [25]. In contrast, our study reported a small number of recurrent VTE cases and a high proportion of distal DVT without PE; therefore, different results may be obtained.

In our study, as well as in previous studies, clear cell carcinoma of the ovary was also a risk factor for VTE [20,21,22]. Four of nine patients with ovarian cancer in the primary outcome had clear cell carcinoma. In previous studies, clear cell carcinoma of the ovary showed clinical features different from those of other epithelial ovarian cancers and was also associated with an increased VTE risk [20,21]. Our study, therefore, indicates that patients with clear cell carcinoma ovarian cancer would be at high risk for a primary outcome, even with treatment by a DOAC.

In line with a previous study, our study revealed that a high D-dimer level when VTE was first diagnosed was a risk factor for VTE recurrence [26]. In contrast, a high fibrin monomer level when VTE was first diagnosed was not a significant risk factor for a primary outcome in our study. However, no previous studies have reported the association between fibrin monomer levels when VTE was first diagnosed and VTE recurrence. The reason for this may be the difference in clinical significance between D-dimer and fibrin monomer. D-dimer levels reflect the condition after thrombus formation, whereas fibrin monomer levels reflect the early stage of VTE onset. Additionally, sensitivity for the diagnosis of thrombosis is thought to decrease when fibrin monomer levels are measured ≥3 days after the onset of symptoms [27,28]. In our study, thrombus was already confirmed in five cases at the first hospital visit; therefore, the difference in clinical relevance between the D-dimer and fibrin monomer levels should be considered when assessing thrombus recurrence [27].

Advanced cancer was also observed in many subjects with a primary outcome in our study. Previous studies reported that advanced stage and hypercoagulation status were associated with VTE onset [29,30]. Within our 2-year observation period, four of the five patients who experienced VTE recurrence had progressive disease or the best supportive care; however, the advanced stage was not significant in our study. This may be due to the small number of research subjects.

To our knowledge, this study was the first to show that ovarian cancer, in particular, clear cell carcinoma of the ovary, PE or proximal DVT without PE when VTE was first diagnosed, and high D-dimer levels when VTE was first diagnosed were risk factors for VTE recurrence/exacerbation or a change from a DOAC to another anticoagulant in patients with gynecologic cancer using DOACs.

However, this study has some limitations. First, this was a single-center study with a small sample size; hence, overfitting in the multiple logistic regression model may exist. Accordingly, a multicenter study with a large sample size is required. Second, the follow-up period was short, and we may have underestimated the primary outcome. Therefore, longer-term investigations are warranted.

Patients with risk factors for a primary outcome require careful follow-up, including detailed interviews and VTE-related physical assessment, venous blood tests (i.e., D-dimer and fibrin monomer levels), and lower limb ultrasonography, after DOAC initiation. Additionally, the CIF results in our study demonstrate that patients with risk factors developed a primary outcome within 2 months after DOAC initiation (Appendix A). Patients with each risk factor, except for clear cell carcinoma of the ovary (i.e., PE or proximal DVT without PE when VTE was first diagnosed, and D-dimer levels in the third tertile (≥7.6 μg/dL) when VTE was first diagnosed), of a primary outcome also had a markedly longer cause-specific RMTL for a primary outcome than those without each risk factor (Appendix A). Therefore, a careful follow-up for patients at high risk for a primary outcome should be considered, especially during the first 2 months after DOAC initiation.

## 5. Conclusions

In conclusion, clear cell carcinoma of the ovary, PE or proximal DVT without PE when VTE was first diagnosed, and D-dimer levels in the third tertile (≥7.6 μg/dL) when VTE was first diagnosed are risk factors for VTE recurrence/exacerbation or a change from a DOAC to another drug within 2 years of VTE diagnosis. Therefore, patients with these risk factors require careful follow-up. When poor thrombus control is suspected, interventions, including switching from a DOAC to another anticoagulant, should be considered to decrease the risk of VTE recurrence/exacerbation.

## Figures and Tables

**Figure 1 cancers-15-01132-f001:**
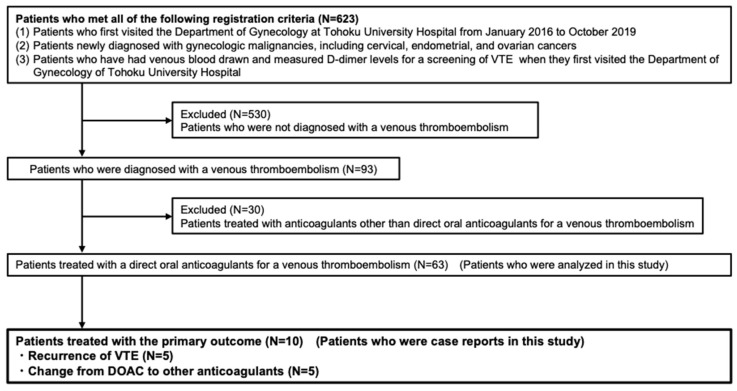
Flowchart of this study.

**Table 1 cancers-15-01132-t001:** Clinical characteristics of the study participants.

Variables	Values (N = 63)
**Age when DOAC was initiated, years**	65.8 (58.6–72.4)
**Age when DOAC was initiated (≥60 years), N (%)**	45 (71.4%)
**Body mass index (kg/m^2^)**	23.5 (20.4–25.5)
**Category of body mass index, N (%)**	
Underweight (<18.5 kg/m^2^)	6 (9.5%)
Normal range (18.5–24.9 kg/m^2^)	40 (63.5%)
Obesity (≥25.0 kg/m^2^)	17 (27.0%)
**Performance status, N (%)**	
0	44 (69.8%)
1	13 (20.6%)
2	6 (9.5%)
**Hypertension, N (%)**	15 (23.8%)
**Diabetes mellitus, N (%)**	5 (7.9%)
**Diagnosis of gynecologic malignancies, N (%)**	
Cervical cancer	8 (12.7%)
Endometrial cancer	21 (33.3%)
Ovarian cancer	34 (54.0%)
**FIGO stage, N (%)**	
Cervical cancer (FIGO stage I/II)	7 (11.1%)
Cervical cancer (FIGO stage III/IV)	1 (1.6%)
Endometrial cancer (FIGO stage I/II)	12 (19.0%)
Endometrial cancer (FIGO stage III/IV)	9 (14.3%)
Ovarian cancer (FIGO stage I/II)	8 (12.7%)
Ovarian cancer (FIGO stage III/IV)	26 (41.3%)
**Pathological diagnosis, N (%)**	
Cervical cancer (SCC)	5 (7.9%)
Cervical cancer (Non-SCC)	3 (4.8%)
Endometrial cancer (Type I)	15 (23.8%)
Endometrial cancer (Type II)	6 (9.5%)
Ovarian cancer (Clear)	9 (14.3%)
Ovarian cancer (Non-clear)	25 (39.7%)
**Location of blood clots when VTE was first diagnosed, N (%)**	
Isolated distal DVT	27 (42.9%)
PE or proximal VTE without PE was present	36 (57.1%)
**Event after initiation of DOAC, N (%)**	
No event	20 (31.7%)
Recurrence of VTE	5 (7.95%)
Change from DOAC to other anticoagulants	5 (7.95%)
Discontinuation of DOAC due to progression of gynecologic malignancies	3 (4.8%)
Discontinuation of DOAC due to bleeding	3 (4.8%)
Discontinuation of DOAC due to the doctor’s discretion	16 (25.4%)
Discontinuation of DOAC due to the subject’s self-judgment	4 (6.3%)
Death from gynecologic malignancies	6 (9.5%)
Death due to other causes	1 (1.6%)

Continuous and categorical variables are shown as median (IQR) and number (percentage), respectively. Abbreviations: DOAC, direct oral anticoagulant; DVT, deep venous thrombosis; FIGO, International Federation of Gynecology and Obstetrics; IQR, interquartile range; PE, pulmonary embolism; SCC, squamous cell carcinoma; VTE, venous thromboembolism

**Table 2 cancers-15-01132-t002:** Laboratory characteristics of the study patients.

Variables	Values (N = 63)
**WBC when VTE was first diagnosed (μ/l)**	6600 (5250–8650)
**Tertiles of WBC, N (%)**	
Tertile 1 (<5600/μL)	22 (34.9%)
Tertile 2 (≥5600 and <8100/μL)	20 (31.7%)
Tertile 3 (≥8100/μL)	21 (33.3%)
**Hemoglobin when VTE was first diagnosed (g/dL)**	10.8 (9.8–12.2)
**Tertiles of Hemoglobin, N (%)**	
Tertile 1 (<10.1 g/dL)	22 (34.9%)
Tertile 2 (≥10.1 and <12.1 g/dL)	21 (33.3%)
Tertile 3 (≥12.1 g/dL)	20 (31.7%)
**Platelet count when VTE was first diagnosed (∗1000/μL)**	308.0 (210.5–396.0)
**Tertiles of Platelet count, N (%)**	
Tertile 1 (<247 ∗ 1000/μL)	21 (33.3%)
Tertile 2 (≥247 and <371 ∗ 1000/μL)	22 (34.9%)
Tertile 3 (≥371 ∗ 1000/μL)	20 (31.7%)
**D-dimer level when VTE was first diagnosed (μg/dL)**	5.1 (2.9–9.0)
**Tertiles of D-dimer, N (%)**	
Tertile 1 (<3.8 μg/dL)	22 (34.9%)
Tertile 2 (≥3.8 and <7.6 μg/dL)	20 (31.7%)
Tertile 3 (≥7.6 μg/dL)	21 (33.3%)
**Fibrin monomer level when VTE was first diagnosed (μg/dL)**	6.4 (3.2–43.9)
**Tertiles of Fibrin monomer, N (%)**	
Tertile 1 (<4.7 μg/dL)	19 (30.2%)
Tertile 2 (≥4.7 and <15.7 μg/dL)	17 (27.0%)
Tertile 3 (≥5.7 μg/dL)	18 (28.6%)
Missing	9 (14.3%)

Continuous and categorical variables are shown as median (IQR) and number (percentage), respectively. VTE, venous thromboembolism; WBC, white blood cell.

**Table 3 cancers-15-01132-t003:** Characteristics of participants with VTE recurrence/exacerbation.

Patient	#1	#2	#3	#4	#5
**Age when DOAC was initiated, years**	73	62	57	57	49
**Body mass index (kg/m^2^)**	19	27.5	24.4	20.5	21.4
**Performance status**	1	2	0	0	0
**Hypertension**	No	No	No	No	No
**Diabetes mellitus**	No	No	No	No	No
**Diagnosis of gynecologic malignancies**	Endometrial cancer	Ovarian cancer	Ovarian cancer	Ovarian cancer	Ovarian cancer
**FIGO stage**	IV	III	I	III	III
**Pathological diagnosis**	Serous carcinoma	Clear cell carcinoma	Endometrioid carcinoma	Clear cell carcinoma	Clear cell carcinoma
**Situation when VTE was first diagnosed**	Before first treatment	Before first treatment	Before first treatment	Before first treatment	Before first treatment
**Location of blood clots when VTE was first diagnosed**	PE, proximal~distal DVT	proximal~distal DVT, Stroke	proximal DVT	PE, distal DVT	PE, distal DVT
**D-dimer level when VTE was first diagnosed (μg/dL)**	36.9	6.6	2.3	8.9	2.8
**Fibrin monomer level when VTE was first diagnosed (μg/dL)**	150	150	-	6	3
**Type of first treatment**	Radiation therapy	Primary debulking surgery	Complete surgery	Primary debulking surgery	Primary debulking surgery
**Type and amount of DOAC prescribed**	Apixaban, 20 mg/day	Apixaban, 10 mg/day	Edoxaban, 30 mg/day	Edoxaban, 30 mg/day	Edoxaban, 30 mg/day
**Duration of DOAC until recurrence/exacerbation of VTE (days)**	5	462	18	18	583
**Situation when VTE recurrence was diagnosed**	After first treatment	After chemotherapy	After first treatment	After first treatment	BSC
**Location of blood clots when VTE recurrence was diagnosed**	Stroke	Stroke	proximal~distal DVT	PE	PE
**RECIST when VTE recurrence/exacerbation was diagnosed**	-	PD	CR	PD	PD
**Cancer-bearing when VTE recurrence was diagnosed**	Yes	Yes	No	Yes	Yes
**Adverse events of DOAC**	No	No	No	No	No
**Final prognosis**	Cause of illness	Cause of illness	Disease-free survival	Cause of illness	Cause of illness

Abbreviations: BSC, best supportive care; CR, complete response; DOAC, direct oral anticoagulant; DVT, deep vein thrombosis; FIGO, International Federation of Gynecology and Obstetrics; PD, progressive disease; PE, pulmonary embolism; PR, partial response; RECIST, response evaluation criteria in solid tumors; SCC, squamous cell carcinoma; SD, stable disease; VTE, venous thromboembolism.

**Table 4 cancers-15-01132-t004:** Characteristics of participants who changed DOACs.

Patient	#1	#2	#3	#4	#5
**Age when DOAC was initiated, years**	70	65	63	61	39
**Body mass index (kg/m^2^)**	24.5	23.4	24	21.3	28.4
**Performance status**	0	0	1	1	0
**Hypertension**	No	No	No	No	No
**Diabetes mellitus**	No	No	No	No	No
**Diagnosis of gynecologic malignancies**	Ovarian cancer	Ovarian cancer	Ovarian cancer	Ovarian cancer	Ovarian cancer
**FIGO stage**	III	III	III	III	IV
**Pathological diagnosis**	Serous carcinoma	Endometrioid carcinoma	Clear cell carcinoma	Serous carcinoma	Serous carcinoma
**Situation when VTE was first diagnosed**	Before first treatment	Before first treatment	Before first treatment	Before first treatment	Before first treatment
**Location of blood clots when VTE was first diagnosed**	PE, proximal~distal DVT	PE, distal DVT	PE, proximal~distal DVT	PE, distal DVT	PE, distal DVT
**D-dimer level when VTE was first diagnosed (μg/dL)**	14.8	17	15.5	34	14.9
**Fibrin monomer level when VTE was first diagnosed (μg/dL)**	150	88.9	5.5	7.8	11.2
**Type of first treatment**	Primary debulking surgery	NAC	NAC	NAC	NAC
**Type and amount of DOAC prescribed**	Rivaroxaban, 30 mg/day	Apixaban, 10 mg/day	Edoxaban, 30 mg/day	Edoxaban, 30 mg/day	Edoxaban, 30 mg/day
**Duration of DOAC until a change from a DOAC to another drug (days)**	3	19	3	60	5
**Situation when a change from a DOAC to another drug**	Before first treatment	Before first treatment	After first treatment	After first treatment	After first treatment
**An increase in D-dimer or fibrin monomer levels when a change from a DOAC to another drug (μg/dL)**	Fib-monomer 5.6 → 33.9	D-dimer 6.1 → 21.1	Fib-monomer 4.2 → 21.8	Fib-monomer 9.1 → 42.9	Fib-monomer 14 → 48
**Days until a change from a DOAC to another drug (days)**	3	19	3	60	5
**Cancer-bearing when a change from a DOAC to another drug**	Yes	Yes	Yes	Yes	Yes
**Adverse events of DOAC**	No	No	No	No	No
**Final prognosis**	Disease-free survival	Cancer-bearing survival	Cause of illness	Disease-free survival	Disease-free survival

Abbreviations: CR, complete response; DOAC, direct oral anticoagulant; DVT, deep vein thrombosis; FIGO, International Federation of Gynecology and Obstetrics; NAC, neoadjuvant chemotherapy; PD, progressive disease; PE, pulmonary embolism; PR, partial response; RECIST, response evaluation criteria in solid tumors; SCC, squamous cell carcinoma; SD, stable disease; VTE, venous thromboembolism.

**Table 5 cancers-15-01132-t005:** Results of the univariate logistic regression model.

Explanatory Variables	Cases/N (%)	Crude OR (95% CI)	*p*-Value
**Age when DOAC was initiated**			
≥60 years	6/39 (13.3)	0.54 (0.13–2.19)	0.4
<60 years	4/14 (22.2)	1.00 (reference)	NA
**BMI ***			
Underweight (<18.5 kg/m^2^)	0/6 (0.0)	0.29 (0.002–2.94)	0.4
Normal range (18.5–24.9 kg/m^2^)	8/40 (20.0)	1.00 (reference)	NA
Obesity (≥25.0 kg/m^2^)	2/17 (11.8)	0.62 (0.11–2.58)	0.5
**Performance status**			
0	6/44 (13.6)	1.00 (reference)	NA
1	3/13 (23.1)	1.90 (0.40–8.96)	0.4
2	1/6 (16.7)	1.27 (0.13–12.8)	0.8
**Diagnosis of gynecologic malignancies ***			
Cervical cancer	0/8 (0)	0.8 (0.005–16.7)	0.9
Endometrial cancer	1/21 (4.8)	1.00 (reference)	NA
Ovarian cancer	9/34 (26.5)	5.1 (1.04–50.3)	0.04
**FIGO stage, all cancers**			
FIGO stage I/II	2/27 (7.4)	1.00 (reference)	NA
FIGO stage III/IV	8/36 (22.2)	3.57 (0.8–25.2)	0.13
**Diagnosis of gynecologic malignancies**			
Clear cell carcinoma	4/9 (44.4)	6.4 (1.34–30.6)	0.02
Ovarian cancer other than clear cell carcinoma or cervical cancer or endometrial cancer	6/54 (11.1)	1.00 (reference)	NA
**Location of blood clots when VTE was first diagnosed ***			
PE or proximal DVT without PE	10/36 (27.8)	21.8 (2.57–2854.6)	0.002
Isolated distal DVT	0/27 (0.0)	1.00 (reference)	NA
**Blood test when VTE was first diagnosed**			
**WBC**			
Tertile 1 or 2 (<8100/μL)	5/42 (11.9)	1.00 (reference)	NA
Tertile 3 (≥8100/μL)	5/21 (23.8)	2.31 (0.59–9.11)	0.23
**Hemoglobin**			
Tertile 1 (<10.1 g/dL)	3/22 (13.6)	1.50 (0.22–10.0)	0.7
Tertile 2 (≥10.1 and <12.1 g/dL)	2/21 (9.5)	1.00 (reference)	NA
Tertile 3 (≥12.1 g/dL)	5/20 (25.0)	3.17 (0.54–18.7)	0.2
**Platelet count**			
Tertile 1 or 2 (<37.1 × 10^4^/μL)	6/43 (14.0)	1.00 (reference)	NA
Tertile 3 (≥37.1 × 10^4^/μL)	4/20 (20.0)	1.54 (0.38–6.22)	0.5
**D-dimer**			
Tertile 1 or 2 (<7.6 μg/dL)	3/42 (7.1)	1.00 (reference)	NA
Tertile 3 (≥7.6 μg/dL)	7/21 (33.3)	6.5 (1.47–28.67)	0.01
**Fibrin monomer ***			
Tertile 1 or 2(<15.7 μg/dL)	5/36 (13.9)	1.00 (reference)	NA
Tertile 3 (≥15.7 μg/dL)	4/18 (22.2)	1.78 (0.42–7.25)	0.42
Missing	1/9 (11.1)	1.01 (0.09–6.12)	0.99

* Firth’s logistic regression model. Abbreviations: BMI, body mass index; CI, confidence interval; DVT, deep vein thrombosis; NA, not applicable; OR, odds ratio; PE, pulmonary embolism; VTE, venous thromboembolism; WBC, white blood cell.

**Table 6 cancers-15-01132-t006:** Results of the multiple logistic regression model.

Explanatory Variables	Adjusted OR (95% CI)	*p*-Value
**Diagnosis of gynecologic malignancies**		
Clear cell carcinoma of the ovary	18.9 (2.25–350.74)	0.005
Ovarian cancer (other than clear cell carcinoma), cervical cancer, or endometrial cancer	1.00 (reference)	NA
**Location of blood clots when VTE was first diagnosed**		
PE or proximal DVT without PE	55.6 (3.29–11,774.66)	0.001
Isolated distal DVT	1.00 (reference)	NA
**D-dimer level when VTE was first diagnosed**		
Tertile 1 or 2 (<7.6 μg/dL)	1.00 (reference)	NA
Tertile 3 (≥7.6 μg/dL)	6.37 (1.17–66.61)	0.03

Firth’s logistic regression model. Abbreviations: NA, not applicable; VTE, venous thromboembolism; OR, odds ratio; PE, pulmonary embolism; DVT, deep vein thrombosis.

## Data Availability

The data presented in this study are available in this article (and Appendix A).

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
