# Peer review of "Precautions during Direct Oral Anticoagulant Introduction in Gynecologic Malignancies: A Single-Center Retrospective Cohort Study"

_cancers, 2023, doi:10.3390/cancers15041132_

Round 1

Reviewer 1 Report

Very interesting study in DOAC for prevention of VTE in GUYN malignancies.

Some questions:

1. How about the guideline compliance of using anticoagulants in prevention of VTE for gynecologic cancer in Japan?

2. 530 patients were excluded due to using anticoagulants other than DOAC for VTE, what kind of other type anticoagulants they used? how about the efficacy of these anticoagulants in prevention of VTE?

3. In section 3.3 participants who changed DOAC, but not due to DOAC adverse events, why they changed the DOAC? what kind of anticoagulant they used?

4. In the study, you have figure out the variables which may affect the timing to recurrent VTE....Do you have any suggestions in prevent these events?

Author Response

Prof.Dr.Samuel C.Mok

Editor-in-Chief

Cancers

Dear Professor Samuel and editors:

Thank you for considering our resubmitted manuscript titled “Precautions during direct oral anticoagulant introduction in gynecologic malignancies: a single-center retrospective cohort study” (manuscript no. cancers-2162769) in Cancers.

We greatly appreciate that the reviewers have recommended publication and would like to thank them for their valuable suggestions. We have revised our manuscript, where appropriate, in accordance with the requested revisions and have outlined our responses to the two comments at the end of this letter.

We hope that our responses and manuscript revisions fully address the reviewers’ concerns. Nonetheless, please do not hesitate to contact us if there are any further questions or concerns.

Yours sincerely,

Hideki Tokunaga, MD, PhD

Department of Obstetrics and Gynecology, Tohoku University Graduate School of Medicine

1-1 Seiryo-machi, Aoba-ku, Sendai, Miyagi 980-8574, Japan

Email: hideki.tokunaga.a1@tohoku.ac.jp

Comments from the Editors and Reviewers:

Reviewer 1

Comments and Suggestions for Authors

Very interesting study in DOAC for prevention of VTE in GUYN malignancies.

RESPONSE: Thank you for the valuable comments. We have revised our manuscript based on your insightful suggestions. We hope the revisions are met with your approval.

  1. How about the guideline compliance of using anticoagulants in prevention of VTE for gynecologic cancer in Japan?

RESPONSE: Thank you for your pertinent comment. We used anticoagulants in reference to the Guidelines for Diagnosis, Treatment, and Prevention of Pulmonary Thromboembolism and Deep Vein Thrombosis (JCS 2017). The corresponding text has been added to the revised manuscript accordingly (p. 3, lines 110-111). 

(p. 3, lines 110-111): Anticoagulants were used in reference to the Japanese guidelines for treatment of VTE.

  1. 530 patients were excluded due to using anticoagulants other than DOAC for VTE, what kind of other type anticoagulants they used? how about the efficacy of these anticoagulants in prevention of VTE?

RESPONSE: Thank you for your question. We have revised the text (p. 4, lines 183-186) to clalify the patients selection process. In brief, 530 patients were excluded because they did not have VTE. We found a misstatement of numbers in Figure 1. We corrected the number of total registerd patients from 817 to 623 in the top panel accordingly.

(p. 4, lines 183-186): Overall, 530 patients who were not initially diagnosed with VTE were excluded. Moreover, 30 patients, who were prescribed anticoagulants other than a DOAC for VTE, including unfractionated heparin or vitamin K antagonists, were excluded.

  1. In section 3.3 participants who changed DOAC, but not due to DOAC adverse events, why they changed the DOAC? what kind of anticoagulant they used?

RESPONSE: Thank you for your valuable comment. The patients whose DOACs were found to be ineffective were given a different drug. The cardiologist or vascular surgen whom we consulted determined that based on the increase of D-dimer or fiblin monomer values. We added the corresponding information to the text to clarify the reason of changing drugs (p.9, lines 251-254). Information regarding the types of DOACs and time to recurrence or exacerbation of VTE has been added to Table 4.

(p.9, lines 251-254): Table 4 shows the clinical characteristics of five patients who changed from DOACs to other anticoagulants according to a cardiologist or vascular surgeon’s decision at our hospital and were deemed as having a poorly-controlled thrombus based on an increase in D-dimer or fibrin monomer levels.

  1. In the study, you have figure out the variables which may affect the timing to recurrent VTE....Do you have any suggestions in prevent these events?

RESPONSE: Thank you for your valuable comment. We showed some risk factors of ineffectiveness of DOACs in Supplemental Table 1. We think it is difficult to prevent recurrence or worsening of VTE; thus, intensive observation and follow-up for patients with the mentioned risk factors is critical. This has been added to the revised manuscript (p13, lines 355-357) accordingly.

 (p13, lines 355-357): Patients with risk factors for a primary outcome require careful follow-up, including detailed interviews and VTE-related physical assessment, venous blood tests (i.e., D-dimer and fibrin monomer levels), and lower limb ultrasonography, after DOAC initiation.

Ultimately, we would like to express our sincere gratitude to the editors and reviewers for their positive and constructive criticism on our manuscript. The manuscript has vastly benefited from your valuable and insightful comments and suggestions. We look forward to hearing from you and would be happy to address any further concerns, if required. We hope this further pushes the manuscript closer to publication in your esteemed journal.

Reviewer 2 Report

This is a well written manuscript describing precautions of oral anticoagulants in gynecological malignancy. The manuscript is of interest to the readers of the journal. The authors may consider the following editorial suggestions:

1. Is it possible to divide Table 1 into 2 tables. I suggest separate table for clinical data and for laboratory variables.

2. It may be of importance to include a list of other drugs administered to patients during this study.

3. It is not clear if these patients are treated with radiation 

4. The dosage of DOACs and duration should be clearly stated

5. The entire manuscript should be checked for consistent usage of abbreviations

Author Response

Prof.Dr.Samuel C.Mok

Editor-in-Chief

Cancers

Dear Professor Samuel and editors:

Thank you for considering our resubmitted manuscript titled “Precautions during direct oral anticoagulant introduction in gynecologic malignancies: a single-center retrospective cohort study” (manuscript no. cancers-2162769) in Cancers.

We greatly appreciate that the reviewers have recommended publication and would like to thank them for their valuable suggestions. We have revised our manuscript, where appropriate, in accordance with the requested revisions and have outlined our responses to the two comments at the end of this letter.

We hope that our responses and manuscript revisions fully address the reviewers’ concerns. Nonetheless, please do not hesitate to contact us if there are any further questions or concerns.

Yours sincerely,

Hideki Tokunaga, MD, PhD

Department of Obstetrics and Gynecology, Tohoku University Graduate School of Medicine

1-1 Seiryo-machi, Aoba-ku, Sendai, Miyagi 980-8574, Japan

Email: hideki.tokunaga.a1@tohoku.ac.jp

Comments from the Editors and Reviewers:

Reviewer 2

Comments and Suggestions for Authors

This is a well written manuscript describing precautions of oral anticoagulants in gynecological malignancy. The manuscript is of interest to the readers of the journal. The authors may consider the following editorial suggestions:

RESPONSE: We appreciate your valuable comments and have modified our manuscript following your kind advice.

  1. Is it possible to divide Table 1 into 2 tables. I suggest separate table for clinical data and for laboratory variables.

RESPONSE: Thank you for your suggestion. We devided table 1 into two tables: Table 1 for clinical and Table 2 for laboratory characteristics.

  1. It may be of importance to include a list of other drugs administered to patients during this study.

RESPONSE: Thank you for your insightful suggestion. We have revised the text accordingly (p. 4, lines 183-186).

(p. 4, lines 183-186): Overall, 530 patients who were not initially diagnosed with VTE were excluded. Moreover, 30 patients, who were prescribed anticoagulants other than a DOAC for VTE, including unfractionated heparin or vitamin K antagonists, were excluded.

  1. It is not clear if these patients are treated with radiation 

RESPONSE: Thank you for your pertinent comment. We added sentences showing our standard treatments for gynecologic malignancies accordingly (p4, lines 188-194).

(p4, lines 188-194): In this study, we followed the Japanese guidelines for treatment of gynecological malignancies. Primary treatments for cervical cancer comprise surgery for early stage and radiation therapy for local advanced tumors. Surgery is the first step in the treatment of endometrial cancer, excluding patients with unresectable aggressive tumor or with poor performance status. In Japan, chemotherapy without radiation represents the most popular adjuvant therapy for endometrial cancer. Table 2 shows laboratory variables of those patients.

  1. The dosage of DOACs and duration should be clearly stated

RESPONSE: Thank you for your pertinent comment. We revised the corresponding tables (new Table 3 and Table 4) to clarify the dosage of DOACs and period of use as per your suggestion.

  1. The entire manuscript should be checked for consistent usage of abbreviations

RESPONSE: Thank you for your suggestion. We have checked the entire manuscript and requested proper editing and revision by a native English speaker. A certification of proofreading by Editage is also attached herewith.

Ultimately, we would like to express our sincere gratitude to the editors and reviewers for their positive and constructive criticism on our manuscript. The manuscript has vastly benefited from your valuable and insightful comments and suggestions. We look forward to hearing from you and would be happy to address any further concerns, if required. We hope this further pushes the manuscript closer to publication in your esteemed journal.

Round 2

Reviewer 1 Report

Thank you very much for your prompt response.